# Laparoscopic Liver Resection Enhanced by an Intervention-Guided Fluorescence Imaging Technique Using Sodium Fluorescein

**DOI:** 10.3390/jcm10163663

**Published:** 2021-08-19

**Authors:** Chang-Min Lee, Min-Young Jeong, Sam-Youl Yoon

**Affiliations:** 1Department of Surgery, Korea University Medical Center Ansan Hospital, Ansan 15355, Korea; laparosamurai@gmail.com; 2Department of Surgery, College of Medicine, Korea University, Seoul 02841, Korea; 3Asan Medical Center, Department of Surgery, College of Medicine, University of Ulsan, Seoul 05535, Korea; nicole0604@naver.com; 4Department of Surgery, Hallym University Sacred Heart Hospital, Anyang 14068, Korea

**Keywords:** laparoscopic liver resection, hepatocellular carcinoma, minimally invasive liver surgery

## Abstract

Background and Objectives: In laparoscopic liver resections, tumor localization is a critical aspect of ensuring clear resection margins and preserving the hepatic parenchyma. In this study, we designed a fluorescence imaging technique using a new fluorophore for tumor localization. Materials and Methods: Immediately before laparoscopic or transthoracic liver resection, microcatheter was inserted through the hepatic artery and used to engrave the segment containing the tumor in the intervention room. Under blue light, the fluorescence of the lesion was visually confirmed, and the location was determined through intraoperative sonography. After tumor localization, liver resection was performed. Results: From February 2017 to March 2020, 24 patients underwent laparoscopic liver resection (LLR) or video-assisted transthoracic liver resection (VTLR) using intervention-guided fluorescence imaging technique (IFIT). Conclusions: IFIT can provide some advantages in the field of LLR. In addition, in cases of VTLR for hepatocellular carcinoma in the superior posterior segment in patients with marginal liver function, IFIT is considered useful.

## 1. Introduction

With the advent of minimally invasive techniques and other medical devices, laparoscopic liver resection (LLR) has been a common procedure since 1992. Once experts in liver surgery established the “Louisville Statement”, a set of guidelines for the rapidly growing area of minimally invasive liver resection [1], the number of reported LLRs has increased consistently. Although minor LLRs have been performed routinely in clinical practice, reports of major and anatomic LLRs have increased sharply [1,2]. Some specialized centers have reported favorable and competitive outcomes of LLR compared to those of open liver resection [3,4,5]. Recently, several reports about single-port LLR, robotic-assisted liver resection, and LLR via video-assisted transthoracic liver resection (VTLR) have been published [6].

In LLR, precise localization of the tumor has been a critical issue based on the need to secure the surgical resection margin and preserve the hepatic parenchyma. In conventional liver resection (CLR), the region of the liver to be resected is determined by using preoperative computed tomography and magnetic resonance imaging. Hepatic segmental resection is carried out after ligation of the Glissonean pedicle; this approach was established to localize liver tumors and uses both the surgeon’s tactile sensations and ultrasound equipment [1,7,8,9]. Although a Glissonean pedicle approach or the tattooing method historically has been carried out to confirm the hepatic resection area in LLR, a Glissonean approach for subdivided branches can be limited by the complicated anatomy of the liver, and laparoscopic indigo carmine tattooing is a technically demanding procedure [10,11]. Recently, several researchers have introduced a near infra-red (NIR) fluorescence imaging technique to create a distinction between liver tumors and their boundaries using indocyanine green (ICG) as a fluorescent tracer during LLR [10,12]. Anatomical resection of the hepatic segment is also achieved with the use of a “negative staining technique,” in which portal branches are closed after the targeted segment is determined [13,14].

However, the tumor-fluorescence technique is limited in identifying superficially located tumors. If a tumor occurs in an area that is not visualized easily in the laparoscopic view, tumor-fluorescence does not help to localize the liver tumor. In such cases, although negative staining technique can be applied in defining the resection range for LLR, it is challenging and time-consuming to expose the portal branches of the targeted hepatic segment. Furthermore, it is more difficult and dangerous to inject any tracer through the arterial or portal branches of the targeted segments than negative staining technique. For these reasons, Ueno et al. proposed an angiographic perfusion method in which ICG is perfused directly through the arterial branch of the targeted segment as a tracer for angiography and visualized with an NIR fluorescence imaging system [11].

We modified this angiographic infusion method to use a different fluorophore in conjunction with preoperative angiography. We were inspired by studies that detailed a blue light fluorescence technique in diverse fields [15,16,17] and therefore adopted sodium fluorescein as a fluorescent tracer to be infused directly into the arterial branches of the targeted segment before LLR or VTLR. In the present study, we report the clinical outcomes of LLR or VTLR in which the tumor-embedding segment was enhanced using an intervention-guided fluorescence imaging technique (IFIT).

## 2. Materials and Methods

This study was approved by the Institutional Review Board of Hallym University Sacred Hospital (registration number: HALLYM 2020-11-003), and the need for individual informed consent was waived by the ethics committee due to the use of anonymized data. All of the procedures were in accordance with the ethical standards of the responsible committees on human experimentation (institutional and national) and with the 1964 Declaration of Helsinki and later versions. Between February 2017 and March 2020, hepatocellular carcinoma (HCC) patients who underwent LLR or VTLR using IFIT were collected retrospectively from a prospectively established database. Surgical procedure.

Immediately before surgery, a 3-Fr sheath catheter (Medikit, Tokyo, Japan) was inserted into the femoral artery in the intervention room, and a 3-Fr catheter (Terumo Corporation, Tokyo, Japan) was placed through the celiac trunk via the aorta. Through the hepatic artery, a 1.9-Fr microcatheter (Terumo Corporation) was used to engrave the segment containing the tumor (Figure 1A,B). After placing a tagging suture to fix the femoral sheath mounted in the inguinal area, the patient was transferred to the operating room and prepared for LLR under general anesthesia (Figure 2A). IMAGE 1 HUB™ HD: (KARL STORZ, Tuttlingen, Germany) was used as an endoscopic system for LLR or VTLR. Liver mobilization was performed to expose the area containing the tumor to be resected. A 10-mm trocar site was used to project the blue light source, and 5 cc of fluorescent tracer was injected via a microcatheter through the femoral artery (Figure 2B). The fluorescent lesion was inspected visually under blue light, and the tumor location was identified using intraoperative sonography to determine whether the tumor was contained sufficiently within the boundaries of the area. Then, after defining the cutting boundary by marking it with monopolar and ultrasonic energy shears, the liver resection was performed using ultrasonic energy shears, a surgical clip, and a monopolar coagulator. During the liver resection, the fluorescent medium remained in the cut liver parenchyma to identify the boundary with the remaining parenchyma.

### 2.1. Detection of Localization under Fluorescent Imaging

We used Dr.’s Light AT (Good Doctors, Incheon, Korea), a commercialized light-emitting diode (LED) curing light that radiates blue light at wavelengths of 440–490 nm as the light source (Figure 3).

In addition, Fluorescite^®^ 10% (Alcon Laboratories, Fort Worth, TX, USA) was used as a fluorescent tracer (Figure 3). This solution contains sodium fluorescein (SF), a fluorophore emitting yellow or green fluorescent light at 520–530 nm when it is stimulated by blue light at wavelengths of 465–490 nm. SF is metabolized in the liver and is eliminated mainly via renal excretion. The renal clearance of fluorescein is estimated to be 1.75 mL/min/kg, while hepatic clearance is 1.50 mL/min/kg.

After administration of undiluted Fluorescite^®^ 10% (Alcon Laboratories) via a microcatheter inserted in the intervention room, the fluorescein-enhanced lesion was detected under blue light emitted from an LED curing light (Figure 4).

### 2.2. VTLR

In cases where CLR, radiofrequency ablation, and transarterial chemoembolization were not possible due to severe cirrhosis, marginal liver function, or diaphragm invasion, liver resection was achieved using a thoracoscopic approach. Single-lung ventilation was required under general anesthesia to perform VTLR. After injection of the fluorescent tracer, the diaphragm adjacent to the area of the fluorescein-enhanced lesion was incised, and 3-0 Vicryl (Ethicon Inc., Raritan, NJ, USA) traction tenting was performed in four to five places (Figure 5A–C). Additionally, the surgeon identified the location of the lesion and the boundary of the segment using intraoperative ultrasound and marked the lesion using a monopolar coagulator; the hepatic parenchyma was cut with ultrasonic energy shears using the Kelly fracture method. The Glissonean pedicle and the hepatic vein seen upon intraoperative ultrasound were ligated using a metal clip when possible. After hepatectomy, the diaphragm was closed with a prolene 3-0 interrupted suture (Figure 5D).

### 2.3. Comparison of Clinicopathologic Outcomes with Those of Internal Controls

The internal control for supplemental analysis was defined as the group of patients who had undergone laparoscopic partial hepatectomy during the same study period. The tumor boundaries in this group consisted mainly of preoperative CT and intraoperative ultrasound. As supplemental analysis, the clinicopathologic outcomes were compared between the internal controls and the patients who underwent IFIT.

## 3. Results

This study included 24 HCC patients who underwent LLR or VTLR from February 2017 to March 2020. Their mean age was 55.3 (49–63) years. The underlying liver disease was hepatitis B in 21 (87.5%) patients, hepatitis C in 1 (4.2%) patient, and alcoholic hepatitis in 2 (8.3%) patients. The mean indocyanine green retention rate at 15 min (ICG 15) of the patient group was 12.4 (8.9–15.2), and the AFP (ng/mL) level was 166 (3.2–200). Preoperative serum INR, total bilirubin, and albumin results were 1.05 (0.89–1.38), 1.03 (0.8–1.3) mg/dL, and 3.88 (3.7–4.2) g/dL, respectively. According to Child–Pugh classification, 20 (83.3%) patients belonged to class A, and 4 (16.7%) were placed in class B. Laparoscopic partial hepatectomy was performed mainly in the right lobe and in liver segment 4. VTLL was carried out in segment 7 in two cases and in segment 8 in two cases (Table 1).

Compared to the internal group, the study group had a significantly reduced average operation time of 221 (143–275) min. The mean time for intervention was 28 (24–31) min. Three (13%) patients required intraoperative blood transfusion. The average amount of blood loss was 200 (10–1100) mL. The mean tumor size was 2.73 (0.70–3.40) cm, and the average distance from the surgical section to the tumor was 1.03 (0.3–2.0) cm; no tumor remained in the cut section in either group. On average, patients were able to consume a normal diet after 2.4 (1–4) days, and were discharged after 10.2(6–14) days of hospitalization (Table 2).

## 4. Discussion

During the last few decades, several studies have investigated the process used to locate tumors during hepatectomy. Currently NIR fluorescence imaging is an innovative way to localize the tumor or anatomical segment to be resected [10,12,13,14,18,19,20]. However, there have been some limitations for this method to be applied in LLR procedures. First, fluorescence imaging can be affected by any biliary excretion disorders that exist in cancerous tissues or in non-cancerous hepatic parenchyma around tumor foci because it is difficult to detect the fluorescence associated with a tumor that is hidden in the inner parenchyma (Figure 1C). To address this issue, surgeons can target the fluorescence in a hepatic segment by injecting ICG into the portal veins or using an intravenous injection of ICG following closure of the proximal portal pedicle toward the hepatic regions to be removed. However, procedures that involve the portal vein or portal pedicles are also difficult during LLR.

Therefore, we designed a new method to create a pathway through which the fluorescent tracer can flow toward the hepatic tumor. Based on prior experiences with TACE, we hypothesized that TACE-associated procedures can be used to achieve fluorescence imaging of hepatic tumors. Since TACE embeds the interventional approach to the artery supplying the hepatic tumor, it is possible to transport the fluorophore to the targeted lesion. This procedure became the main strategy for IFIT.

Ueno et al. [11] also reported LLR using IFIT, in which ICG-mixed gelatin embolization and NIR fluorescence imaging were performed to visualize the tumor lesion. However, although ICG is used as a fluorescent tracer retained in the liver parenchyma [11], NIR fluorescence using ICG has shown some barriers to standardization.

Compared with the conventional laparoscopic surgical systems, NIR-associated laparoscopic surgical systems are expensive and, therefore, not available at all medical facilities. When NIR fluorescence imaging initially was introduced, its fluorescence-detecting system could visualize only the excited fluorophore; therefore, surgeons should disable the NIR mode to view the surrounding structures. In other words, due to the longer wavelengths of NIR than the visible spectrum, the fluorescence and surrounding structures were not visualized simultaneously [15]. This issue might be a significant drawback for hepato-biliary surgery where most procedures involve critical structures [16]. More recently, technologic advances have emerged to solve this issue by providing fusion imaging of NIR and visible light. However, this fusion imaging system has a high cost that includes the tremendous load required to manage the fusion images as well as the economic barrier. We suggest that this is why IFIT using NIR has not been disseminated widely. IFIT requires preoperative intervention, and the additional equipment required might be another hindrance to standardization.

Therefore, instead of NIR, we adopted blue light fluorescence. This modality does not require any separate device other than a conventional laparoscopic or thoracoscopic system because blue light fluorescence is visible to the naked eye. That is, IFIT using blue light fluorescence has the advantage of being able to derive possible results using only fluorescein and a blue light generator. Fluorescein has been used widely for retinography in the ophthalmic area and has been applied in several studies showing the feasibility of blue light fluorescence [15,16,17]. In addition, the blue light source is easier to prepare than an NIR fluorescence-associated fusion imaging system.

In this study, we applied IFIT using blue light fluorescence during LLR or VTLR. In conventional laparoscopic partial hepatectomy, it takes a lot of time to find small areas of HCC located in the dome or round ligament of the liver using intraoperative ultrasound. Moreover, even when resecting the liver, the need to assess continually the boundary using repeated intraoperative ultrasound is burdensome and causes a delay in the operation. However, IFIT using blue light fluorescence brightly illuminates the region to be resected, reducing the time required to find a tumor. Even surgeons who experience difficulty using laparoscopic ultrasound intraoperatively can secure a margin of safety and identify surrounding boundaries if the ultrasound is performed after tumor localization using the IFIT method. Although a distorted liver surface can occur after liver mobilization, the marker generated by IFIT remains in place and can help to determine the tumor location.

IFIT was useful in VTLR to find tumors located in the superior posterior segment. Tumors in this segment are difficult to access laparoscopically; even if LLR is performed, excess parenchyma might be removed. If patients with marginal liver function have lesions in the superior posterior segment, VTLR can be considered. However, the surgical view provided by thoracoscopic access to the liver is not familiar to both thoracic surgeons and hepatobiliary surgeons [2,3,8,21]. In addition, another problem with VTLR is that, unlike LLR, the movement of the diaphragm creates an unstable surgical view. Therefore, it is important to determine the exact location of the liver tumor and to perform minimal liver resection. In such case, if the IFIT method is used, hepatic resection can be performed through a small diaphragmatic incision after the exact location of the liver tumor is determined [6]. In this study, it was possible to locate rapidly the hepatic tumor that fluoresces under the diaphragm, reducing operation time. Even after a tumor is located, it is important to determine the boundary because it is difficult to grasp the structure of the liver as seen through the incised diaphragm. In this case, intraoperative laparoscopic ultrasound can help determine the location and boundaries of a liver tumor.

One limitation of the IFIT procedure is the possibility of complications related to the femoral catheter. Such complications did not occur in this study, but the risks of this technique are similar to those of general embolization. In addition, attention should be taken to prevent catheter-related infection, bleeding and thrombosis of the femoral artery when catheters are placed before surgery. The use of a catheter should be restricted in patients with intestinal arteriovenous fistulas [11].

In conclusion, IFIT using blue light fluorescence can provide a useful guidance to define the resection ranges during LLR. Additionally, in patients with marginal liver function where VTLR is performed to treat HCC located in the superior posterior segment, IFIT is useful. Although no catheter-related complications have been reported, anticoagulant treatment and rapid catheter removal after injection of the fluorescent contrast agent should be considered to reduce or prevent possible complications.

## Figures and Tables

**Figure 1 jcm-10-03663-f001:**
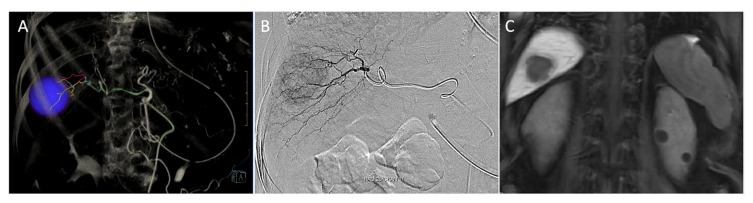
Preoperative angiography and magnetic resonance imaging of hepatocellular carcinoma (HCC). (**A**) Tumor localization (blue circle) and feeding artery simulation on 3D angiography CT. (**B**) A 1.9-Fr microcatheter was placed through the hepatic artery to engrave the segment containing the tumor under angiography. (**C**) Magnetic resonance imaging showed a HCC located in segment 7 with a longest diameter of 3.1 cm.

**Figure 2 jcm-10-03663-f002:**
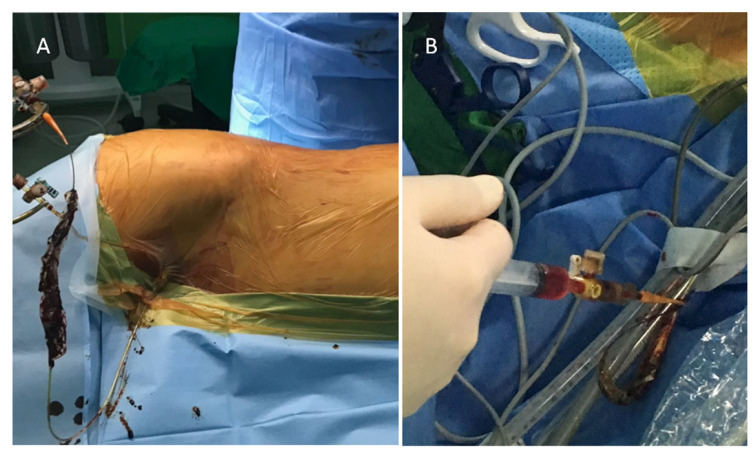
Administration of fluorophore via microcatheter. (**A**) The microcatheter was fixed with tagging sutures on the inguinal area. (**B**) 5 cc of undiluted Fluorescite 10% was injected via the microcatheter through the femoral artery.

**Figure 3 jcm-10-03663-f003:**
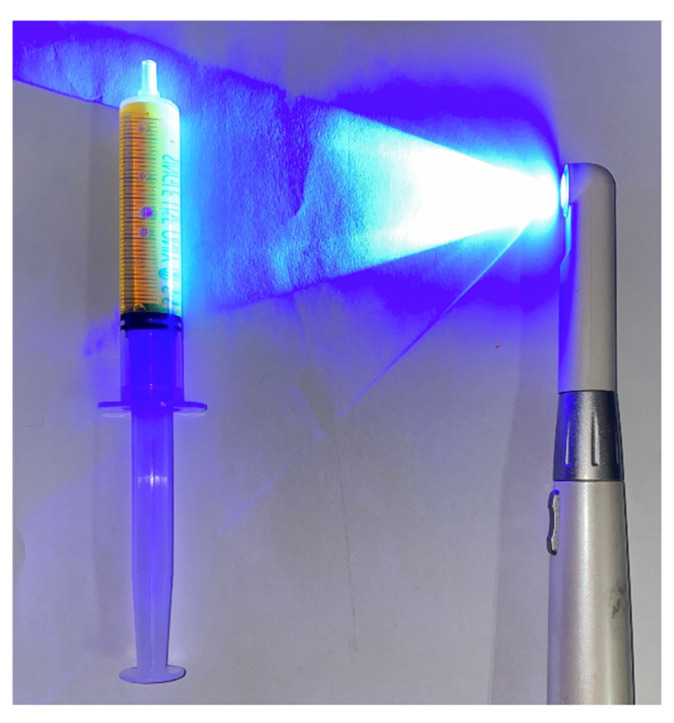
The light source. A commercialized curing light was used as a light source for blue light fluorescence.

**Figure 4 jcm-10-03663-f004:**
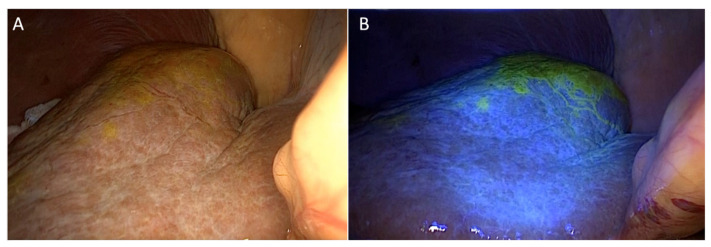
Laparoscopic liver resection using intervention-guided fluorescence imaging technique. (**A**) No fluorescence appeared on the liver surface without blue light. (**B**) Hepatic segment 7 fluoresced under blue light (same case as shown in Figure 1).

**Figure 5 jcm-10-03663-f005:**
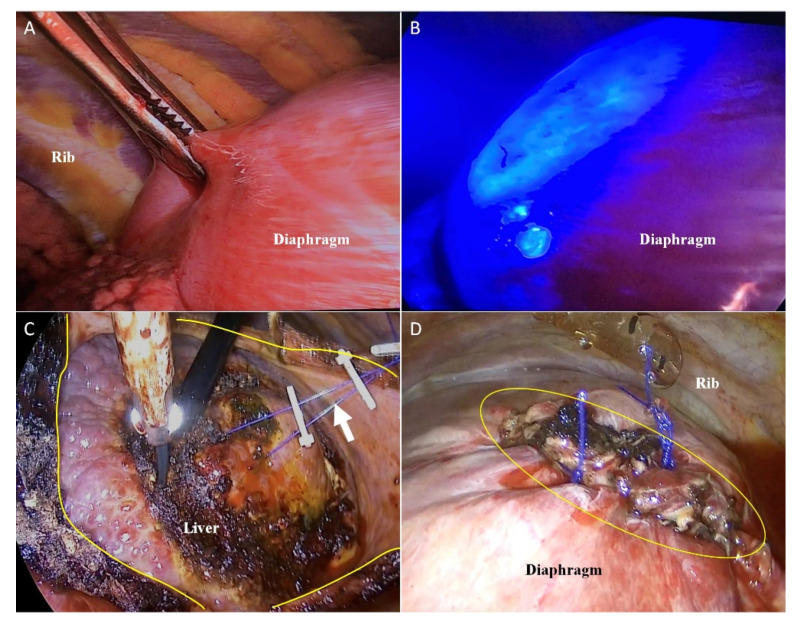
Video-assisted transthoracic liver resection (**A**) Without blue light, there were no indications of tumor location on thoracoscopic view. (**B**) Tumor fluorescence from segment 7 was illuminated through the diaphragm. (**C**) The tumor-bearing region was resected using ultrasonic energy shears (yellow line: incision line on the diaphragm; white arrow: tenting suture). (**D**) The incised diaphragm was closed using non-absorbable suture materials.

**Table 1 jcm-10-03663-t001:** Clinicopathologic data of the patients enrolled in the current study.

Patients Who Underwent Intervention-Guided Fluorescence Imaging Technique (*n* = 24)	
Age	55.3 (49–63)
Sex ratio (Male: Female)	2:1
Liver disease	
Hepatitis B	21 (87.5%)
Hepatitis C	1 (4.2%)
Alcoholic hepatitis	2 (8.3%)
ICG 15 (%)	12.4 (8.9–15.2)
AFP (ng/mL)	166 (3.2–200)
Platelets, ×10^3^/mm^3^	143 (121–182)
INR	1.05 (0.89–1.38)
Total bilirubin (mg/dl)	1.03 (0.8–1.3)
Albumin (g/dL)	3.88 (3.7–4.2)
CTP score	
A	20 (83.3%)
B	4 (16.7%)
Tumor location	
IV	6 (6 LLR)
V	5 (5 LLR)
VI	4 (4 LLR)
VII	5 (3 LLR, 2 VTLR)
VIII	4 (2 LLR, 2 VTLR)

ICG 15 indicates indocyanine green retention rate at 15 min; AFP, α-fetoprotein; INR, International Normalized Ratio; CTP score, Child–Turcotte–Pugh score for severity of liver cirrhosis; LLR, laparoscopic liver resection; VTLR, Video-assisted transthoracic liver resection.

**Table 2 jcm-10-03663-t002:** Comparison of the clinicopathologic outcomes between the study group and internal controls.

	IFIT (*n* = 24)	Internal Controls (*n* = 29)	*p*
Operation time (min)	221 (143–275)	265 (200–300)	<0.001
Time to the first semi-fluid diet (days)	2.4 (1–4)	2.8 (1–5)	0.222
Transfusion ^a^	3 (13%)	4 (13.8%)	0.758
Blood loss (cc)	200 (10–1100)	215 (5–1300)	0.438
Hospital stay (days)	10.2 (6–14)	10.0 (6–15)	0.556
Resection margin (cm)	1.03 (0.3–2.0)	1.01(0.2–3.0)	0.587
Tumor size	2.73 (0.70–3.40)	2.51 (0.5–3.5)	0.412

^a^ Number of patients who underwent perioperative transfusion. IFIT, the patients who underwent intervention-guided fluorescence imaging technique.

## Data Availability

The data presented in this study are available on request from the corresponding author.

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
