# Peer review of "Laparoscopic Liver Resection Enhanced by an Intervention-Guided Fluorescence Imaging Technique Using Sodium Fluorescein"

_jcm, 2021, doi:10.3390/jcm10163663_

Round 1

Reviewer 1 Report

Congratulations for your work. However, your results and data cannot reach to conclusion. 

  1. Usually, ICG-guide surgery can be done by infusion of ICG through peripheral vein. But, transarterial approach is too invasive and I cannot find any additional advantage.
  2. According to your figures, tumor detection looks to be not so dramatic.
  3. Many surgery was done for superficially located tumor or visually detectable tumors. I cannot find any improvement in terms of tumor detection.
  4. Comparison with internal group is not so attractive because both groups are not homogeneous and not comparative.

Author Response

Congratulations for your work. However, your results and data cannot reach to conclusion.

1) Usually, ICG-guide surgery can be done by infusion of ICG through peripheral vein. But, transarterial approach is too invasive and I cannot find any additional advantage.

 Yes, thanks for your sincere comment. As you commented, ICG-guide surgery can be done by infusion of ICG through peripheral vein. For example, ICG-guide surgery for hepatocellular carcinoma (HCC) is possible by peripheral infusion of ICG, if the tumor is superficially located in the liver. However, if the tumor is deeply located in the liver, we cannot make a fluorescence on the segment by peripheral infusion. Regarding this issue, although Ishizawa et al. tried to make a fluorescence imaging on a specific segment by infusion of ICG through portal branch, it is difficult and dangerous to inject any tracer through the arterial or portal branches of the targeted hepatic segments during laparoscopic surgery. That is the reason why Ueno et al. designed transarterial approach for ICG-infusion (Ueno M et al. Indocyanine green fluorescence imaging techniques and interventional radiology during laparoscopic anatomical liver resection. Surgical endoscopy 2018;32:1051-55). Our transarterial approach is more advantageous than Ueno et al.’ technique, because we use fluorescein (another fluorescent dye) instead of ICG. It is necessary to use a specific equipment for visualizing ICG-fluorescence, since ICG-fluorescence is out of the visual spectrum (we cannot see this through the bare eyes.). However, we can see fluorescein-fluorescence through the bare eyes, because fluorescein-fluorescence is inside the visual spectrum. This benefit is correlated with the standardization of this procedure, as the necessity of a specific equipment can be a big hindrance to the standardization of the procedure.

2) According to your figures, tumor detection looks to be not so dramatic.

 Yes, thanks for your comment. At this time, we tried to get the dramatic figures from the operation video, and therefore we submitted the better figures than before.

3) Many surgery was done for superficially located tumor or visually detectable tumors. I cannot find any improvement in terms of tumor detection.

 Yes, thank you for your sincere comment. Our technique is more significant in the cases of the deeply located tumors. All the cases undergoing IFIT are the deeply-located tumors. In addition, the figures also include the case of the deeply located tumor. To show that we applied IFIT in the case of the deeply located tumor, we added the magnetic resonance image in Figure 1.

4) Comparison with internal group is not so attractive because both groups are not homogeneous and not comparative.

 Yes, thanks for your comment. We agree with your comment. This study is retrospective, and therefore the internal group cannot be a perfect one for the comparative analysis. To our best knowledge, this is the first trial for Intervention-guided Fluorescence Imaging Technique (IFIT) using fluorescein, and therefore it is necessary to accomplish a pilot study to be a basis for the prospective study. In this study, we did not intend to show the superiority of IFIT (using fluorescein) to the conventional surgery. Insteads, we tend to show the non-inferiority of our technique to the conventional surgery. However, in the next phase of our study, we have a plan to show the superiority of IFIT using fluorescein.

Reviewer 2 Report

The authors showed that intervention-guided fluorescence imaging using sodium fluorescein could provide visualization of tumor-bearing hepatic region during laparoscopic liver resection (LLR).

Currently, ICG fluorescence imaging have been widely adopted during LLR and anatomical liver resection can be performed by positive staining & negative staining techniques. There are a lot of publication about ICG fluorescence, but this report is very unique in the respects of using a new fluorescein and intervention technology.

However, the authors should solve these problems in below. 

  1. The authors should elucidate the function of Fluorescite, including the  metabolizm and clearance. Is it excreted into the bile duct? 
  2. Is Fluorescite commercialized? Is it approved use or off-label use for the venous and arterial infusion?
  3. Can this fluorescence be visualized normal laparoscopic scope? The author should show the name of laparoscopic scope. How did they use the light source during laparoscopic operation?
  4. In the figure 2, 3, and 5, the case presentation including location, diameter, etc. should be shown precisely. The readers cannot understand what procedures were performed.
  5. What is the indication of the IFIT although they compared with conventional LLR in the same period. The LLR seemed to be partial liver resection in this study.
  6. I consider this technique is not necessary for partial resection because it is easy to remove and IFIT takes the excessive invasiveness and cost. The author should explain it.  
  7. The operation time was significantly shorten by IFIT, but the authors should show the interventional time as well. And how did the authors decide which artery to be embolized?
  8. How do the authors think about generalization and standardization of this technology?

Author Response

There are a lot of publication about ICG fluorescence, but this report is very unique in the respects of using a new fluorescein and intervention technology.

However, the authors should solve these problems in below.

1) The authors should elucidate the function of Fluorescite, including the  metabolizm and clearance. Is it excreted into the bile duct?

 Yes, thank you for your sincere comment. We used 10% FLUORESCITE® (Alcon Laboratories, TX, USA) as the fluorescent tracer. This solution contains sodium fluorescein; This is metabolized in the liver and is mainly eliminated via renal excretion. The renal clearance of fluorescein is estimated at 1.75 ml/min/kg, while hepatic clearance is 1.50 ml/min/kg. Systemic clearance is essentially complete within 48–72 h after administration of 500 mg sodium fluorescein. We will add these contents in ‘Materials & methods’ section.

2) Is Fluorescite commercialized? Is it approved use or off-label use for the venous and arterial infusion?

 Yes,thanks for your comment. Fluorescite is commercialized. In this study, we used 10% FLUORESCITE® (Alcon Laboratories, TX, USA). This dye has been used as an imaging agent for retinal angiography since 1960. Therefore, it had already been approved for the vascular infusion. Furthermore, there are several literatures regarding the use of fluorescein in the other fields; (1) T. Ichikawa et al. Intra-Arterial Fluorescence Angiography with Injection of Fluorescein Sodium from the Superficial Temporal Artery during Aneurysm Surgery: Technical Notes. Neurol Med Chir (Tokyo). 2014 Jun; 54(6): 490–496., (2) K. Kuroda et al. Intra-arterial injection fluorescein videoangiography in aneurysm surgery. Neurosurgery. 2013 Jun;72. We described this content in ‘Discussion’ section.

3) Can this fluorescence be visualized normal laparoscopic scope? The author should show the name of laparoscopic scope. How did they use the light source during laparoscopic operation?

 Yes, thank your for the significant comment. The most important benefit of our technique is the fluorescence can be visualized by the normal laparoscopic scope. According to your request, we will describe the name of our laparoscopic scope in ‘Materials & methods’ section. In addition, we will describe how we use the light source during laparoscopic operation in ‘Materials & methods’ section; a 10-mm trocar site wound was used to project the blue light source, as the wound size is appropriate for inserting the light source.

4) In the figure 2, 3, and 5, the case presentation including location, diameter, etc. should be shown precisely. The readers cannot understand what procedures were performed.

 Yes, thank you. We absolutely agree with your sincere comment. We will describe the case presentation including the tumor location, diameter, et al. as the Legends of the figures.

5) What is the indication of the IFIT although they compared with conventional LLR in the same period. The LLR seemed to be partial liver resection in this study.

 Yes, thanks for your comment. The most important indication of the IFIT is the agreements of the patients, because it might seem to be time-consuming or complicated. Therefore, in the same period, some patients did not agree with the IFIT, and therefore the conventional operation should be performed in these cases (internal controls). In this study, it is important to clarify the resection-range for LLR, in which we performed partial liver resection.

6) I consider this technique is not necessary for partial resection because it is easy to remove and IFIT takes the excessive invasiveness and cost. The author should explain it.

 Yes, thank you for the significant comment. We also think that the expert surgeons can perform partial resection. However, we designed our technique (IFIT using fluorescein) as an easy way for the novice surgeons to perform the partial resection. For the surgeons who perform a few LLR in low-volume center, it is not easy to clarify the resection-range, regardless of the tumor location. Furthermore, when they meet the tumor located deeply in the liver parenchyme, it is very difficult to perform laparoscopic approach.

7) The operation time was significantly shorten by IFIT, but the authors should show the interventional time as well. And how did the authors decide which artery to be embolized?

 Yes, thank you. It is an important comment. We will show the interventional time in ‘Results’ section. In addition, we decided which artery to be infused with fluorescite according to the images of the computed tomography (CT) angiography (Figure 1).

8) How do the authors think about generalization and standardization of this technology?

 Thanks for your sincere comment. IFIT using fluorescein can be easily generalized and standardized by the following reasons; first, our IFIT procedure has been originated from transarterial chemoembolization (TACE). TACE has been performed widely in the secondary or tertiary hospital, and therefore transarterial infusion of fluorescite can be widely accepted in many institutes. Second, fluorescence using ICG can be visualized by the specific system for detecting NIR fluorescence, and therefore the generalization and standardization can be limited by the requirement for the specific system (furthermore, this system is still expensive and not reimbursed in the most country.). Here, we should remark that our technique (IFIT using fluorescein) does not require such specific system; fluorescence using fluorescein can be visualized by the ordinary laparoscopic scope. Thus, IFIT using fluorescein can be generalized and standardized.

Round 2

Reviewer 1 Report

All my queries were well responded. However, in title, "endoscopic" should be changed to "laparoscopic".

Author Response

Reviewer1: All my queries were well responded. However, in title, "endoscopic" should be changed to "laparoscopic".

à Yes, thanks for your sincere comenet. The reason why we use the word “endoscopic” was that we applied transthoracic approach in several cases as well as laparoscopic approach. However, according to your sincere recommendation, we will change “endoscopic” into “laparoscopic” in title.

Reviewer 2 Report

Thank you for your revision. The authors’ fluorescence technique is interesting. but now there are a lot of reports about anatomical liver resection using ICG fluorescence imaging and they can perform LLR for tumors in S7/8 without thoracoscopic approach even if tumor location is not good. I think that interventional technique is excessive, so this value is questionable.     

Author Response

Reviewer2: Thank you for your revision. The authors’ fluorescence technique is interesting. but now there are a lot of reports about anatomical liver resection using ICG fluorescence imaging and they can perform LLR for tumors in S7/8 without thoracoscopic approach even if tumor location is not good. I think that interventional technique is excessive, so this value is questionable.

à Yes, thank you for the sincere comment. As you commented, there are some reports about “anatomical” liver resection using ICG fluorescence imaging in which perform LLR for tumors in S7/8 without thoracoscopic approach. However, the targets of our technique were the cases we should perform “non-anatomical” resection, including severe liver cirrhosis, marginal liver function, or diaphragm invasion. In such cases, we comtemplated that “non-anatomical” resection was more appropriate than “anatomical resection”. Therefore, If patients with marginal liver function have lesions in the superior posterior segment, we adopted thoracoscopic approach to facilitate the minimal resection (e.i. wedge resection). We described this contents in “Materials & method” and “Discussion” sections.